# The Apparent Organ-Specificity of Amyloidogenic ApoA-I Variants Is Linked to Tissue-Specific Extracellular Matrix Components

**DOI:** 10.3390/ijms24010318

**Published:** 2022-12-24

**Authors:** Rita Del Giudice, Mikaela Lindvall, Oktawia Nilsson, Daria Maria Monti, Jens O. Lagerstedt

**Affiliations:** 1Department of Experimental Medical Science, Lund University, 221 84 Lund, Sweden; 2Department of Chemical Sciences, University of Napoli Federico II, Complesso Universitario Monte Sant’Angelo, 80126 Napoli, Italy; 3Istituto Nazionale di Biostrutture e Biosistemi (INBB), 00136 Rome, Italy; 4Islet Cell Exocytosis, Lund University Diabetes Centre, Department of Clinical Sciences Malmö, Lund University, 20506 Malmö, Sweden

**Keywords:** Apolipoprotein A-I, cytotoxicity, extracellular matrix components, amyloidosis

## Abstract

Apolipoprotein A-I (ApoA-I) amyloidosis is a rare protein misfolding disease where fibrils of the N-terminal domain of the protein accumulate in several organs, leading to their failure. Although ApoA-I amyloidosis is systemic, the different amyloidogenic variants show a preferential tissue accumulation that appears to correlate with the location of the mutation in the protein sequence and with the local extracellular microenvironment. However, the factors leading to cell/tissues damage, as well as the mechanisms behind the observed organ specificity are mostly unknown. Therefore, we investigated the impact of ApoA-I variants on cell physiology and the mechanisms driving the observed tissue specificity. We focused on four ApoA-I amyloidogenic variants and analyzed their cytotoxicity as well as their ability to alter redox homeostasis in cell lines from different tissues (liver, kidney, heart, skin). Moreover, variant-specific interactions with extracellular matrix (ECM) components were measured by synchrotron radiation circular dichroism and enzyme-linked immunosorbent assay. Data indicated that ApoA-I variants exerted a cytotoxic effect in a time and cell-type-specific manner that seems to be due to protein accumulation in lysosomes. Interestingly, the ApoA-I variants exhibited specific preferential binding to the ECM components, reflecting their tissue accumulation pattern *in vivo*. While the binding did not to appear to affect protein conformations in solution, extended incubation of the amyloidogenic variants in the presence of different ECM components resulted in different aggregation propensity and aggregation patterns.

## 1. Introduction

Amyloid diseases, or protein misfolding diseases, are caused by inherent instability in specific proteins and peptides that aggregate into insoluble fibrils which accumulate in tissues and organs, ultimately causing organ failure [1]. So far, about 40 unrelated proteins have been found to be responsible of human amyloid pathologies. Only in a few cases these proteins will accumulate in the cytosol whereas, in most cases, amyloidogenic proteins give rise to fibrils accumulating in the extracellular space [2]. This is also the case of variants of apolipoprotein A-I (ApoA-I).

Albeit being mostly known for its health-promoting functions in atherosclerosis and diabetes [3,4,5,6], ApoA-I is also responsible for a rare form of systemic late-onset, hereditary amyloidosis. This disease is caused by twenty-three of the many naturally occurring mutations in the *APOA1* gene [7,8,9,10,11], and these amyloidogenic mutations (mainly substitutions but also deletions and frameshifts) all lead to the production of variant forms of the protein endowed with lower affinity for lipids and enhanced ability to undertake the fibrillogenic pathway [12,13]. The ApoA-I amyloidogenic mutations cluster in two major hot spots of the protein sequence that include the residues 26–107 and 154–178, which all give rise to morphologically identical amyloid deposits, constituted by the 90–110 amino acid long N-terminal fragment of the variant protein [14].

Currently, no therapeutic approach is available to cure ApoA-I related amyloidosis. The only effective option is liver transplantation to reduce amyloid production and slow down the progression of the disease, and transplantation of the failing organs [15]. A deeper knowledge of the molecular mechanism of ApoA-I amyloidosis is thus required to develop effective therapeutic tools aimed at preventing fibrils accumulation, or to reduce and/or clear amyloid deposits in the affected organs. However, despite the many efforts devoted to the understanding of the connection between the altered protein structure-function of ApoA-I amyloidogenic variants and the clinical phenotype [13,16], there is still much that needs to be uncovered about the molecular mechanisms underlying this genetic disorder. Factors leading to the destabilization of the interaction between the ApoA-I variants and the lipids as well as the processes responsible for damage of cells/tissues are largely unknown. Another intriguing, yet still not understood, aspect of this pathology is that, while ApoA-I amyloidosis is a systemic disease, the different amyloidogenic variants show a preferential tissue accumulation that clearly appears to be dependent on the location of the amino acid substitutions and on their distinctive interactions with tissue-specific extracellular milieus. Indeed, hepatic and renal amyloidosis are typical for the amino acid substitutions in the region 1 to 75, whereas cardiac, laryngeal and cutaneous depositions have been reported for ApoA-I variants of the region 170 to 178 [9,17].

To shed light on these intriguing and unresolved aspects in ApoA-I amyloidosis, we focused on the ApoA-I amyloidogenic variants G26R and L75P, accumulating mainly in liver and kidney, and L174S and L178H, forming fibrils preferentially in heart and skin [18,19,20,21]. Here, we studied their effect on target cells and their interaction with tissue-specific components of the extracellular matrix (ECM), in comparison to the native protein. Our data show that ApoA-I amyloidogenic variants exert a selective, cell-specific, and time-dependent cytotoxic effect on the cell lines tested. Moreover, the aggregation patterns of the different variants and their affinities for the ECM components are in agreement with the clinically described tissue preferences. The findings here reported provide a deeper mechanistic understanding on how a specific amyloidogenic mutation in the ApoA-I protein dictates its tissue specificity. The knowledge gained will be fundamental to develop drugs able to prevent or control this rare disease.

## 2. Results

### 2.1. The Effects of ApoA-I Amyloidogenic Variants on Target Cells Are Variant and Cell-Type Specific

The effects of ApoA-I amyloidogenic variants on cell lines from different ectopic origins were analyzed to investigate the disparities between the variants and the WT protein on cell physiology. Cell lines were chosen to represent tissues in which the ApoA-I amyloidogenic variants are known to accumulate as amyloid deposits. Rat kidney epithelial cells (NRK-52E) and human hepatocytes (HepG2) were used to represent kidney and liver, respectively, where G26R and L75P preferential accumulate *in vivo*. Human keratinocytes (HaCaT) and rat cardiomyoblasts (H9c2) were chosen to represent skin and heart, respectively, targets for the preferential aggregation of L174S and L178H.

#### 2.1.1. ApoA-I Amyloidogenic Variants Are Cytotoxic in a Cell-Specific Manner

First, the effect of lipid-free ApoA-I amyloidogenic variants on cell viability was assessed. Cells were incubated for different lengths of time (24, 48 and 72 h) with the four recombinant ApoA-I amyloidogenic variants (lipid-free), as well as with the wild-type protein, at 0.5, 1 and 2 μM, and cell viability was evaluated by MTT assay. The recombinant proteins exerted a time- and dose-dependent cytotoxic effect on all the cell lines tested (Appendix A, Figure 1). The data reported in Figure 1 show the results obtained at 2 μM, which approximates the physiologic concentration of lipid-free ApoA-I in plasma. Intriguingly, these data show that G26R and L75P variants have a selective toxicity on kidney and liver cells (right panels), whereas L174S and L178H variants, were found to be more toxic on skin- and heart-derived cells (left panels). Moreover, data showed that H9c2 and HaCaT cell viability was affected to a higher extent compared to the other two cell lines tested. Indeed, the detrimental effect of the L174S and L178H variants on cell viability (purple and orange bars, respectively, in Figure 1) could be observed already after 24 h and 48 h for H9c2 and HaCaT, respectively, compared to the 72 h of incubation needed to observe effects of G26R and L75P (red and green bars, respectively, in Figure 1) on NRK-52E and HepG2 viability.

These results suggest that the ApoA-I amyloidogenic variants exert a cytotoxic effect in a selective and cell-specific manner that agrees with the variants’ preferential tissue specificities observed in the clinical studies.

Cell viability upon treatment with reconstituted HDL (rHDL) particles containing the different ApoA-I variants was also tested (Appendix A). However, the negative effects of the different variants on cell viability were significantly attenuated, and a variant-specific cytotoxic effect was observed only after 72 h of incubation on the HaCat and NRK-52E cells. These results indicated that lipidation mitigates the toxicity exerted by ApoA-I amyloidogenic variants on target cells.

#### 2.1.2. Cell-Specific Impairment of Degradation of ApoA-I Amyloidogenic Variants

A growing body of evidence shows that mitochondrial and lysosomal dysfunctions contribute to the detrimental effects of amyloid diseases on target organs [22,23]. Thus, following the treatment of cells with the different ApoA-I variants, the cellular redox state and the cellular fate of the recombinant ApoA-I proteins were analyzed.

To understand a potential role of redox homeostasis imbalance for the observed cytotoxic effect, cellular Reactive Oxygen Species (ROS) levels were assessed by DCFDA assay. Cells were incubated with either recombinant lipid-free WT ApoA-I or one of the four ApoA-I amyloidogenic variants (lipid-free) for different length of time (6, 24, 48 or 72 h), and Antimycin A or TBHP as positive controls. Appendix A shows that none of the ApoA-I amyloidogenic variants exogenously added to the cells were able to induce a significant alteration of the intracellular ROS, indicating that the observed cytotoxic effect is not due to an imbalance in the redox status.

Next, the uptake and intracellular levels of the ApoA-I variants in target cells were evaluated by confocal microscopy analysis on all the four cell lines.

Cells were treated for different length of time with FITC-labelled ApoA-I proteins, and lysosomes were stained with lysotracker red. Confocal microscopy on H9c2 cells, reported in Figure 2, shows that, after 6 h of incubation, all the ApoA-I protein variants, as well as WT protein, were partially internalized, as observed by an increase in the cytosolic content of labeled ApoA-I proteins. However, the signal intensities of the FITC-labelled L174S and L178H variants at 6 h appeared higher compared to the other variants and to the WT protein. Moreover, the signal associated with the L174S variant partially co-compartmentalized with lysosomes (white arrows in Figure 2, upper panel). These observations indicate that either the uptake of these two variants is more efficient or that their cellular degradation is partially impaired. To explore this further, cells were incubated with ApoA-I variant proteins for 6 h followed by a 16 h wash-out period (i.e., incubation of cells in the absence of ApoA-I protein). The observation that cells incubated with the L174S and L178H variants still presented a strong protein signal that co-compartmentalize with lysosomes (indicated by the white arrows in Figure 2, middle panel) supports the hypothesis of impaired lysosomal protein degradation. Indeed, cells treated with the WT protein, as well as with the other two amyloidogenic ApoA-I proteins (G26R and L75P), showed only a faint protein signal indicating that these proteins were efficiently degraded during the wash-out period. As a control, cells were incubated for 24 h in the presence of the different labelled protein variants (Figure 2, lower panels). All five ApoA-I proteins were associated with increased cytosolic signals which partially co-compartmentalized with lysosomes. This is in agreement with previous reports on the WT protein indicating that lysosomes contribute to the partial degradation of the ApoA-I proteins but also constitute a compartment for intracellular storage of ApoA-I [24].

The results obtained on NRK-52E kidney epithelial cells are presented in Figure 3. As in the case of cardiomyoblasts, all the tested ApoA-I proteins tested were efficiently internalized by the kidney epithelial cells during the 6 h incubation, with an apparent higher fluorescent signal for the G26R and L75P variants (Figure 3, upper panels). Incubation of the cells with the ApoA-I proteins for 6 h followed by 16 h washout of the FITC-labelled proteins (Figure 3, middle panels) resulted in re-localization of all five ApoA-I protein variants to the lysosomal compartment. However, a lower fluorescent signal was observed for the WT protein and the L174S and L178H variants, as compared to the fluorescent signal of the G26R and L75P variants, indicating lysosomal degradation of these ApoA-I proteins. Conversely, the apparent accumulation of the G26R and L75P variants in the lysosomes suggests a partial impairment of the lysosomal activity. Finally, as observed in cardiomyoblasts (Figure 2), 24 h incubation with the protein variants lead to lysosomal accumulation for all ApoA-I proteins (Figure 3, lower panels).

Similar results were obtained on HaCaT keratinocytes (Appendix A) and HepG2 hepatocytes (Appendix A), where the apparent accumulation of the variants in the cellular compartment and the partial impairment of the lysosomal activity followed the clinical observed tissue-specificity of the amyloidogenic variants.

### 2.2. ApoA-I Amyloidogenic Variants Exhibit Specific Preferential Binding to Different ECM Components

The results presented so far indicate that ApoA-I amyloidogenic variants exert variant- and cell-type specific effects on the physiology of target cells. The reason for this specificity might reside in the interaction between the different ApoA-I variants and specific components of the cell membrane and/or the extracellular matrix (ECM) that characterize an organ or a cell-type. Therefore, the ability of ApoA-I amyloidogenic variants to bind to different ECM components characteristic of different extracellular environments was tested. To this purpose, enzyme-linked immunosorbent (ELISA) assays were performed by coating plates with the ECM component under test (heparin, heparin sulfate, dermatan sulfate, hyaluronate and chondroitin sulfate) and probing with increasing concentrations of lipid-free recombinant ApoA-I amyloidogenic variants, or WT ApoA-I as reference. As shown in Figure 4, all the variants showed both an increased affinity (lower K_d_) and a higher capacity (higher B_max_) to bind heparin, compared to the WT protein. This general increase in the ability to bind to heparin has previously been shown for other amyloidogenic proteins [25,26]. As described below, there was a variant specific binding to the other ECM components (heparin sulfate, dermatan sulfate, hyaluronate and chondroitin sulfate).

The L174S and L178H variants were characterized by a higher capacity to bind dermatan sulfate and hyaluronate, as indicated by higher B_max_ values compared to wild-type ApoA-I. In addition, L178H showed a higher affinity also for heparan sulfate and chondroitin sulfate. Notably, these two ECM components are particularly abundant in skin [27], which is one of the target tissues for L174S and L178H variants amyloidogenic variants. On the contrary, L75P and G26R showed higher affinity for heparan sulfate and chondroitin sulfate, which are abundant also in liver and kidney [27]. Of these, L75P was found to have a higher affinity for chondroitin sulfate, whereas G26R, although characterized by a low capacity to bind heparan sulfate, showed a significantly higher affinity for this ECM component compared to the other variants and the WT protein. These results are in good agreement with ApoA-I variants preferential tissue accumulation and with the ectopic localization of the selected ECM components, suggesting that the extracellular environment contributes to the different types of clinical manifestations of ApoA-I amyloidosis.

The ability of rHDL containing either the WT, G26R or L174S ApoA-I proteins to bind heparin was tested by ELISA assay. The results reported in Figure 5 show that also in the lipid bound form, the variants retain a higher capacity to bind heparin compared to the WT.

The comparison of the lipid-bound ApoA-I proteins to their lipid-free counterpart indicated that lipidation greatly reduces the extent of the binding of the proteins to heparin, especially in the case of the WT protein (Appendix A). Indeed, although the affinity (K_d_) of WT protein for heparin did not significantly change upon protein lipidation, this protein showed a higher decrease in heparin binding capacity compared to the amyloidogenic variants (97.5% reduction in B_max_ for the WT vs. 83.8 and 77.6%, for G26R and L174S, respectively).

### 2.3. ApoA-I Amyloidogenic Variants Undergo Different Conformational Changes in the Presence of Different ECM Components

The impact of the different ECM components on the conformation of the ApoA-I amyloidogenic variants was inspected by high-throughput synchrotron radiation circular dichroism (HT-SRCD). For this purpose, ApoA-I amyloidogenic proteins as well as the WT protein were incubated with the different ECM components at physiological pH, at 37 °C for up to 36 h. Samples were then analyzed by HT-SRCD, and the secondary structure composition of the ApoA-I proteins calculated.

In the time frame of the experiment (36 h), no significant changes in the secondary structure composition of the different ApoA-I proteins could be captured. Yet, incubations with the ECM components led to trends for secondary structural changes (Appendix A). These included: (i) heparin-induced increase of β-strand at the expenses of unordered structure in the WT protein; (ii) reduction of α-helix and concomitant increase of β-structures in the presence of heparin, heparan sulfate and dermatan sulfate in the case of L75P, and in the presence of heparan sulfate, dermatan sulfate and chondroitin sulfate for the L174S; (iii) drop in α-helical structure along with increase in unordered following the addition of each of the ECM components, excluding heparan sulfate, in the L178H samples. Samples incubated for time periods longer than 36 h were also analyzed, however, at these later time points significant amounts of the proteins had aggregated which prohibited comparative in-solution SRCD analyses.

In order to mimic a pathophysiological environment and to promote amyloid aggregation of ApoA-I variants, we performed experiments at pH 6.4, which corresponds to the pH of inflamed tissues. ApoA-I proteins were incubated in the absence or presence of either heparin, heparan sulfate or hyaluronate at pH 6.4 for 1 week at 37 °C. The ECM components included in this experiment were chosen based on the results presented in Figure 4 and were the components for which the ApoA-I variants showed either an altered binding affinity or capacity compared to the WT. At the end of the incubation the supernatant and insoluble fractions were separated and analyzed by SRCD.

The SRCD spectra obtained on the soluble and insoluble material, reported in Appendix A, were used for the secondary structure estimation shown in Figure 6. As expected, for all the ApoA-I proteins, the incubation at mild acidic pH, in absence of any ECM component, induced the formation of a conspicuous protein pellet characterized by protein with secondary structure composition shifted towards a higher fraction of β-strands, as compared to the protein in the soluble fraction, indicating pH-induced fibrillar aggregation (Figure 6, lower panels). For the soluble proteins in the supernatant, the structure of the WT protein underwent negligible changes, whereas the fraction of β-strand structure increased considerably for all the variants (especially for the L75P variant) at expenses of the α-helix content, both in absence and in presence of any ECM components (Figure 6, upper panels). Some key differences could be detected among the variants and the different ECM components added. The soluble G26R variant presented a very similar secondary structure regardless of the ECM components present, except for a higher α-helix percentage in the presence of heparan sulfate. The soluble L75P and L178H variants also showed no significant change in structure between the different conditions. However, the soluble fraction of the L174S protein incubated with ECM component presented very clear differences in structure in the presence of heparan sulfate and hyaluronate, showing a significant higher contribution of the α-helical structure compared to control, accompanied by a reduced β-strand structure.

The secondary structure patterns of the different ApoA-I proteins in the insoluble protein pellets showed distinct behavior. The WT ApoA-I presented a significant decrease in α-helix along with an increase in β-strands, compared to the control (no addition), when incubated in the presence of heparin, in line with previous reports [28,29]. The same behavior could be observed for the insoluble L75P variant in the presence of heparin. The insoluble G26R variant, instead, presented a significantly larger fraction of α-helix in the presence of both heparin and hyaluronate. A clear increase in α-helix fraction could also be observed for the insoluble L174S variant incubated in the presence of heparan sulfate and hyaluronate, same conditions at which the soluble protein presented α-helical increase at the expenses of the β-strand fraction. Finally, the insoluble L178H variant presented a very similar secondary structure compositions in all the conditions tested.

## 3. Discussion

ApoA-I amyloidosis is a degenerative disease for which no cure is currently available. Indeed, the only way to slow down the progression of the disease is the transplantation of the organs affected or of the liver to reduce the production of the mutated ApoA-I protein [15]. The search for any therapeutic strategy requires a deep knowledge of the molecular mechanisms of the disease, thus, the elucidation of determinants of protein instability and aggregation, along with the identification of the mechanisms responsible for tissue derangement in ApoA-I amyloidosis are of pivotal importance.

To gain knowledge on this, we selected four ApoA-I amyloidogenic variants, two of which accumulate mainly in kidney and liver (G26R and L75P), and two leading to the deposition of fibrils mostly in heart and skin (L174S and L178H) [18,19,20,21], and analyzed their effects on target cells as well as their interactions with selected components of the extracellular matrix.

The deposition and progressive accumulation of protein fibrils in target organs has been long considered as the main pathogenic agent that leads to the failure of the organ. Indeed, amyloid deposits can be particularly massive [1] and the steric rigidity of amyloid fibrils [30] could also contribute to the physical distortion of the affected tissue, thus compromising organ function [1]. However, more recent studies indicate that amyloidogenic proteins can also exert a cytotoxic activity when in their oligomeric, prefibrillar forms, as in the case of transthyretin [31,32], amyloid β peptide (Aβ) [33], and the N-terminal domain of ApoA-I [34]. In this study we demonstrated that the full-length ApoA-I amyloidogenic variants lead to toxic effect on target cells. Furthermore, we demonstrated that the different ApoA-I amyloidogenic variants can exert a time dependent and cell-type specific toxicity (Figure 1), thus recapitulating clinical findings, and that this toxic effect seems to be due to protein accumulation in the lysosomes (Figure 2 and Figure 3). The confocal microscopy images revealed a high protein accumulation in the cell compartment of cardiomyoblasts already after 6 h incubation with L174S and L178H ApoA-I variants, and the fluorescent signal associated with the proteins compartmentalizes with lysosomes also after 16 h incubation in the protein-free culture medium (wash out). This is particularly evident in cells incubated with the L174S variant (Figure 2), indicating that lysosomes are not efficiently clearing the accumulating protein. As a consequence, the viability of cardiomyoblasts incubated with L174S and L178H ApoA-I variants was found to be significantly compromised at 24 h (Figure 1, upper-left panel).

We observed a different effect of the ApoA-I variants on renal cells. Although a high fluorescent signal for the G26R and L75P proteins is observed in renal cells after 6 h incubation followed by 16 h in protein-free culture medium, we do not observe a strong protein accumulation after 24 h (Figure 3). This might indicate that the renal cells are able to overcome the toxic protein accumulation, at least for short incubation times. This agrees with the observation that, in renal cells, the cytotoxic effect of the variants is visible only upon longer incubation times (72 h) with the G26R and L75P variants (Figure 1, upper-right panel).

The effects of ApoA-I amyloidogenic variants on target cells might be due to the misfolding-prone structure of the variants, which makes them readily redirected to the lysosomes for degradation. Here, the acidic environment of lysosomes could also contribute to the aggregation of the variants, thereby making the proteins less accessible to proteolysis. This hypothesis would also explain the previously observed partial intracellular accumulation of the L75P variant in HepG2 cells [35], and is corroborated by the findings from Jayaraman and colleagues who demonstrated that serum amyloid A (SAA) is redirected to lysosomes and forms stable proteolysis-resistant soluble oligomers at the lysosomal acidic pH [36].

Lysosomal dysfunction in amyloid diseases is often accompanied by mitochondrial impairment and is thought to contribute to the detrimental effects of amyloids on target organs [22,23]. Incubation of the different cell lines with the four amyloidogenic variants did not result in unbalanced redox homeostasis (Appendix A). This is in contrast with our recent report on hepatic cells stably transfected with the L75P ApoA-I amyloidogenic variant that showed adverse effect of the mutated protein on redox homeostasis and mitochondria functionality [37]. This indicates that the endogenously expressed L75P ApoA-I variant follows a different cellular fate compared to that of the exogenously added protein. This is of relevance since hepatic cells are the major producers of ApoA-I protein in circulation, hence all other tissue types will be affected by ApoA-I protein endocytosed from blood circulation.

This study also pinpoints the importance of the role of the extracellular environment for the observed ApoA-I variant- and cell-type specific toxic effect. Indeed, since the extracellular amyloid deposition in vivo takes place in a heterogeneous environment, it is conceivable that the interaction of amyloidogenic proteins with other biomolecules, such as components of the cell membrane and/or the extracellular matrix, may dramatically affect their structural organization, shifting the equilibrium from a native protein conformation toward a pathological state. Interestingly, published reports demonstrated that specific cellular environments can trigger the aggregation of L75P and L174S ApoA-I amyloidogenic variants in a selective way [38]. Among the extracellular matrix components, glycosaminoglycans (GAGs) provide differentiated biological environments in mammals; heparin and heparan sulfate are the most ubiquitous GAGs at the cell surface and in the extracellular matrix, whereas chondroitin sulfate is mostly found in many connective tissues and skin [27].

We demonstrated that the ApoA-I variants display binding patterns to the different ECM components that reflect the observed tissue accumulation pattern (Figure 4). The binding appeared not to affect protein conformations initially, however extended incubation of the ApoA-I variants in the presence of different ECM components at pathophysiological pH yielded different aggregation propensities and aggregation patterns (Figure 6 and Appendix A). In the analysis of the observed aggregates, it is important to consider that amorphous aggregates might co-exist with the fibrillar species in the pellet fraction. This might explain why some of the protein in the pellet fraction still showed a considerable percentage of α-helix. Further analyses will be needed to study the conformation of the protein in the insoluble material to fully decipher the nature of the aggregates.

The results agree with reports showing that GAGs play an active role in promoting amyloid fibril formation and stabilization [39,40]. Furthermore, among GAGs, heparin and heparin sulfate have been found to have a role in a variety of amyloid disorders including Alzheimer’s disease, type II diabetes, light chain amyloidosis, and prion-related diseases [25,26]. Of relevance, heparin has been reported to enhance the amyloid aggregation of WT ApoA-I *in vitro*, especially in oxidative environment and at acidic pH [28,29].

We also demonstrated that lipidation plays a fundamental role in the pathogenesis of ApoA-I amyloidosis. Indeed, lipidation attenuates the cytotoxic effects of the amyloidogenic variants (Appendix A) and reduces the level of protein binding to heparin, especially in the case of the native protein (Figure 5 and Appendix A). Despite the generally higher stability of the lipidated proteins, the variants retain a higher capacity to bind heparin compared to the native protein, suggesting that heparin has a destabilizing effect on the lipid-protein interaction. The interaction between ApoA-I and lipids has already been shown to be weaker in HDL containing ApoA-I amyloidogenic variants. Indeed, in our previous studies we demonstrated that the variants have a lower affinity for phospholipids [13] and that the equilibrium between lipid-free and bound states is slightly shifted towards the lipid-free form [16]. The presence of heparin, or of any other specific ECM component, might promote the dissociation of the variant from the HDL, thus shifting the equilibrium between lipid-free/bound forms further towards the lipid-free state, promoting protein misfolding and aggregation. Our findings are strengthened by the observation that heparin promotes the simultaneous dissociation of SAA and ApoA-I from HDL-SAA particles [41] leading to the formation of aggregates of the two proteins with heparin.

## 4. Materials and Methods

### 4.1. Protein Expression and Purification

Human ApoA-I proteins, containing at the N-terminus a 6-Histidine-tag and tobacco etch virus (TEV) protease recognition site, were produced in *Escherichia coli* BL21(DE3) pLysS (Invitrogen, Thermo Fisher Scientific, Waltham, MA, USA) as described in [42].

The recombinant proteins were isolated using an immobilized metal affinity chromatography (IMAC, His-Trap-Nickel-chelating columns, GE Healthcare, Chicago, IL, USA) and the his-tag removed by TEV protease treatment. The his-tag free proteins were then purified by performing a second IMAC [43]. Protein purity was analyzed by SDS-PAGE, followed by Coomassie staining, and protein concentration was determined by using a NanoDrop 2000c spectrophotometer (Thermo Fisher Scientific).

### 4.2. Preparation of Reconstituted HDL

Reconstituted HDL (rHDL) particles were produced as described in [44]. Briefly, a dry DMPC (1,2-dimyristoyl-sn-glycero-3-phosphocholine, Avanti Polar Lipids, Alabaster, AL, USA) film was solubilized in PBS, and multilamellar vesicles generated extruding the lipid solution through a 100 nm polycarbonate membrane with the LiposoFast system (Avestin, Ottawa, ON, Canada). rHDLs were obtained by incubating the recombinant ApoA-I proteins with DMPC multilamellar vesicles at 1:100 protein to lipid molar ratio and a protein concentration of 2 mg/mL at 24 °C, for 72 h. To isolate homogeneous 9.6 nm rHDL, size-exclusion chromatography was employed by using a Superose 6 increase 10/300 GL column (GE Healthcare) and samples were eluted in PBS at a flow rate of 0.5 mL/min.

### 4.3. Cell Culture

Human hepatocytes (HepG2, product number HB-8065), rat cardiomyoblasts (H9c2, product number CRL-1446,) and rat kidney epithelial cells (NRK-52E, product number CRL-1571) were purchased from American Type Culture Collection (ATCC, Manassas, VA, USA), whereas human keratinocytes (HaCat) were obtained from Prof Ole Sorensen (Lund University, Sweden) [45].

Cells were cultured in DMEM with Glutamax (Gibco) supplemented with 1% antibiotic (penicillin, streptomycin; Invitrogen) and 5% calf serum (Gibco, Waltham, MA, USA) for NRK-52E cells or 10% fetal bovine serum (HyClone; Thermo Scientific) for the other cell lines, in a 5% CO_2_ humidified atmosphere at 37 °C. For H9c2 cells, DMEM containing 1 mM sodium pyruvate (ATCC) was used.

### 4.4. Cytotoxicity Assessment

HepG2 and H9c2 cells were seeded in 96-well plates at 3000 cells/well while HaCaT and NRK-52E cells at a density of 2000 cells/well. After 24 h incubation at 37 °C, cells were treated with each ApoA-I protein, either in the lipid-free (at 0.5, 1 or 2 μM) or lipid bound form (2 μM), for 24, 48 and 72 h. At the end of the incubation, a 3-(4,5-dimethylthiazol-2-yl)-2,5-diphenyltetrazolium bromide (MTT) assay was performed for the assessment of cell viability. MTT reagent (Sigma-Aldrich, Burlington, MA, USA) was solubilized in DMEM without phenol red (Sigma-Aldrich) and added to the cells to a final concentration of 0.5 mg/mL. Following a 4 h incubation at 37 °C, the culture medium was removed and the resulting formazan salts dissolved with 0.1 N HCl in isopropanol. Absorbance values were measured at 570 nm using an automatic multiplate reader (Thermo Scientific-Multiskan Go). Cell survival was expressed as the percentage of viable cells in the presence of the protein under test with respect to control cells grown in the absence of the protein.

### 4.5. Reactive Oxygen Species (ROS) Production

HepG2 and H9c2 cells were seeded at a density of 2000 cells/well whereas HaCaT and NRK-52E cells at 1000 cells/well in 96-well plates and incubated at 37 °C for 24 h prior to the treatment with 2 μM of lipid-free ApoA-I proteins, for 6, 24, 48 and 72 h. 1 mM Antimycin A or 50 μM tert-Butyl hydroperoxide (TBHP) were used as positive controls for the formation of ROS in HepG2 and NRK-52E or H9c2 and HaCat, respectively. One hour prior to the end of the incubation, 20 μM 2′,7′-dichlorodihydrofluorescein diacetate (DCFDA) was added to the cells and the plates incubated at 37 °C for additional 45 min. At the end of the incubation, ROS production was assessed by measuring the dichlorodihydrofluorescein (DCF) fluorescence by exciting at 485 nm and following the emission at 535 nm, using an automatic multiplate reader (FLUOstar^®^ Omega, BMG Labtech, Ortenberg, Germany). ROS production was expressed as the percentage of DCF fluorescence in the presence of the protein under test compared to non-treated cells.

### 4.6. Protein Labelling and Confocal Microscopy

ApoA-I proteins, purified as described above, were dialyzed in 0.1 M sodium carbonate at pH 9.0, and conjugated to fluorescein isothiocyanate (FITC), according to the manufacturer instructions (Sigma-Aldrich). FITC-labelled ApoA-I proteins were separated from the unbound FITC by using PD-10 desalting columns containing Sephadex G-25 (GE-Healthcare), equilibrated in PBS.

Cells were seeded on glass cover slips in 24-well plates and grown to semi-confluency. Then, after incubation of the cells with 2 μM FITC-ApoA-I proteins in culture medium for 6 or 24 h, lysosomes were stained with 75 nM LysoTracker Red (Invitrogen) for 1 h at 37 °C, and nuclei were stained with Hoechst 33342 (Invitrogen), for 15 min at 37 °C. Cells were then extensively washed with PBS and fixed with 4% paraformaldehyde in PBS for 15 min at room temperature. Slides were rinsed with water and mounted on 5 μL of Fluoromount Aqueous Mounting Medium (Sigma). Samples were visualized using a Nikon Confocal A1+ microscope, equipped with 488 nm, 561 nm and 647 nm lasers, using 60x objective. Image formatting and analysis was performed using NikonA1+ NIS Elements software and ImageJ software, version 2.0.0-rc-59/1.51j.

### 4.7. Enzyme-Linked Immunosorbent Assay (ELISA)

96-well plates were coated with 25 μg/mL of each of the five ECM components under test diluted in PBS. After overnight incubation at room temperature under agitation, plates were washed with PBS and the free surface was blocked with 1% BSA in PBS for 1 h at 37 °C. The plates were then probed with the different ApoA-I proteins, at increasing concentrations (0–100 μg/mL range) for 2 h at 37 °C. After washing with PBS, samples were incubated with biotinylated anti-human ApoA-I antibodies (Abcam ab27630) for 1 h at 37 °C and thereafter with alkaline phosphatase (ExtrAvidin-AP, Sigma) for 30 min at 37 °C. The assay was developed using p-nitrophenyl phosphate (SIGMAFAST™p-Nitrophenyl phosphate tablets, Sigma) and detected by reading the absorbance at 450 nm using an automatic multiplate reader (Thermo Scientific-Multiskan Go).

### 4.8. Synchrotron Radiation Circular Dichroism (SRCD) on ApoA-I Amyloidogenic Variants Incubated in the Presence of Different ECM Components

SRCD experiments were performed at the B23 Beamline at the Diamond Light Source by using a nitrogen-flushed Module A and B end-station spectrophotometer [46,47]. All the spectra were acquired at 25 °C in the far-UV range 185–260 nm, with a 1 nm wavelength increment, and were corrected subtracting the background signal of the buffer. Estimation of secondary structure from CD spectra was performed on CDApps [48] using the Continll algorithm with reference data SP 43 [49]. The molar ellipticity ([Θ]) was calculated according to the equation described in [50].

#### 4.8.1. Analysis at Physiological pH—High-Throughput SRCD

ApoA-I protein samples were dialyzed against McIlvaine buffer (165 mM Na_2_HPO_4_, 17.6 mM citrate, pH 7) to minimize the signal-to-noise ratio at wavelengths below 200 nm. Protein samples were diluted to 0.5 mg/mL, mixed with each ECM components at a 1:2 protein to ECM component molar ratio, and then incubated at 37 °C with 350 rpm agitation, for different lengths of time. At the indicated time points, protein samples were loaded into the quartz 96 multi-well plate [51] (0.2 mm path length), and the spectra were acquired as described above.

#### 4.8.2. Analysis at Pathophysiological pH

ApoA-I amyloidogenic variants as well as the native protein (2 mg/mL), dialyzed against McIlvain buffer, pH 6.4, were incubated with the different ECM components (1:2 protein to ECM ratio) for one week at 37 °C, under agitation. At the end of the incubation, samples were centrifuged at 20,000× *g* for 30 min. Protein concentration in the supernatant was estimated by use of a NanoDrop 2000c spectrophotometer. The amount of protein in the pellet was determined by subtracting the amount of protein in the supernatant at the end of the incubation from the amount of protein at the beginning. Aliquots of the supernatants were placed in a 0.1 mm quartz cuvette without further dilution and analyzed by SRCD as described above. For analysis of the pellet fractions, each sample was thoroughly resuspended in buffer to a concentration of 2 mg/mL, then 5 µL of the solution were spread on a quartz slide over an approximately 1 cm^2^ area and dried under a flow of nitrogen gas. The slide was then placed in the spectrometer and the SRCD spectra acquired.

### 4.9. Statistical Analysis

Data shown are the mean ± SD or ± SEM, as indicated. Analysis was performed either by one-way ANOVA with Dunnett’s post hoc test or by two-way ANOVA with Tukey’s post hoc test, as indicated in figure legends, using the GraphPad Prism 8 software, version 8.4.3.

## 5. Conclusions

The results described in this study provide a possible link between the apparent organ specificity of ApoA-I amyloidogenic variants, their effects on target cells, and tissue specific distribution of GAGs. The mechanism that we propose, depicted in Figure 7, arises from the observation that the equilibrium between lipid-free and HDL-bound forms is shifted towards the less structurally stable lipid-free state for the ApoA-I amyloidogenic variants.

Components of the extracellular matrix, and particularly tissue-specific GAGs, might further promote the dissociation of the ApoA-I variants from the HDL particles, with two potential consequences. On one hand, the interaction of the ApoA-I variants with the tissue-specific GAGs promotes fibrillar protein aggregation, eventually leading to amyloid accumulation in the target tissues. On the other hand, this interaction facilitates the internalization of the ApoA-I variants in specific cell types. Here, the misfolded variants are readily directed to the lysosomes for degradation, however, the acidic lysosomal environment might promote protein aggregation and the subsequent accumulation of the variants. The consequential lysosomal dysfunction results in cell death and, eventually, in the disruption of the tissue architecture.

## Figures and Tables

**Figure 1 ijms-24-00318-f001:**
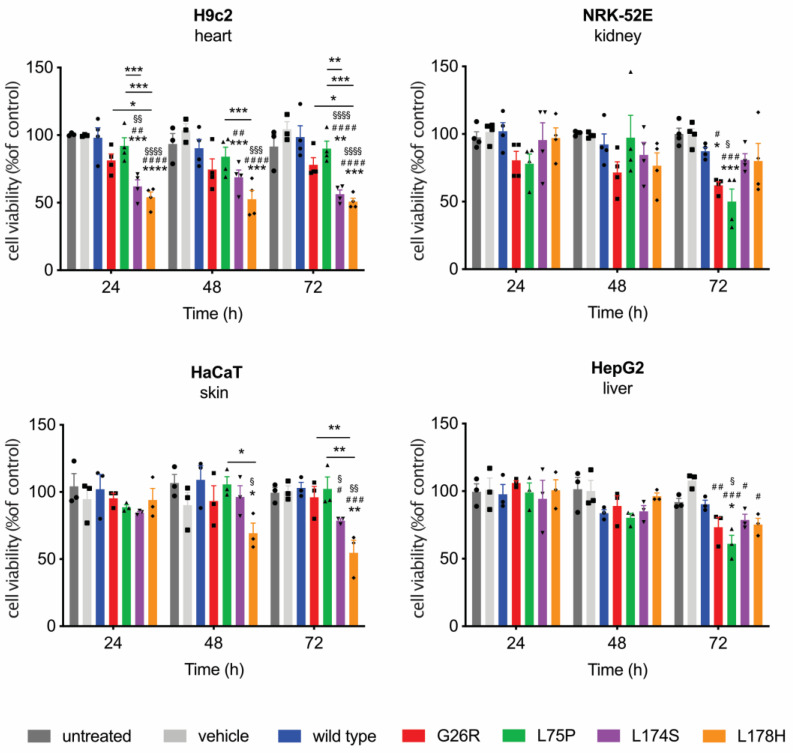
ApoA-I amyloidogenic variants affect cell viability in a cell-type specific way. H9c2, NRK-52E, HaCaT and HepG2 cells were incubated with 2 μM of each ApoA-I amyloidogenic variant, as well as the WT protein, for the indicated length of time. At the end of incubation, cell viability was determined by performing a MTT (3-[4,5-dimethylthiazol-2-yl]-2,5-diphenyltetrazolium bromide) assay. Data shown are the means ± SEM of at least three independent experiments carried out in triplicate. Significance was calculated according to 2way ANOVA (* *p* < 0.05, ** *p* < 0.005, *** *p* < 0.001, **** *p* < 0.0001 refers to groups as indicated with respect to untreated cells, # *p* < 0.05, ## *p* < 0.005, ### *p* < 0.001, #### *p* < 0.0001 refers to groups as indicated with respect to vehicle treated cells, § *p* < 0.05, §§ *p* < 0.005, §§§ *p* < 0.001, §§§§ *p* < 0.0001 refers to groups as indicated with respect to the WT protein).

**Figure 2 ijms-24-00318-f002:**
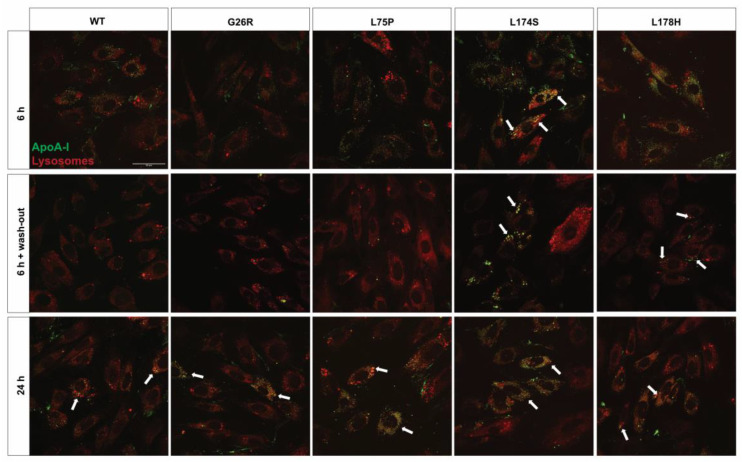
Cellular localization of ApoA-I amyloidogenic variants in H9c2 cardiomyoblasts. H9c2 cells were incubated at 37 °C with 2 µM FITC-labelled ApoA-I proteins (green) for 6 h (upper and middle panels) or 24 h (lower panels). To study the ability of the lysosomes to degrade the ApoA-I proteins, cells incubated with FITC-ApoA-I for 6 h were incubated for additional 16 h in absence of the labelled protein (6 h + wash-out, middle panel). Lysosomes were stained with Lysotracker red. Cells were imaged by confocal microscopy using a 60× objective. Scale bar is 50 μm. White arrows indicate protein co-compartmentalization with the lysosomes.

**Figure 3 ijms-24-00318-f003:**
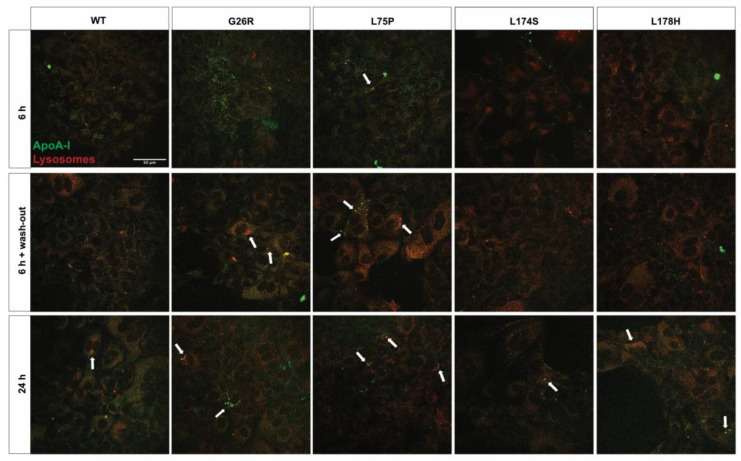
Cellular localization of ApoA-I amyloidogenic variants in NRK-52E kidney cells. NRK-52E cells were incubated with 2 μM FITC-labelled ApoA-I (green) for 6 h (upper panel), for 6 h followed by incubation in growth medium for additional 16 h to study protein degradation (6 h + wash-out, middle panel), or 24 h (lower panel). Lysosomes were stained with Lysotracker red. Cells were analyzed by confocal microscopy using a 60× objective. Scale bar is 50 μm. White arrows indicate protein co-compartmentalization with the lysosomes.

**Figure 4 ijms-24-00318-f004:**
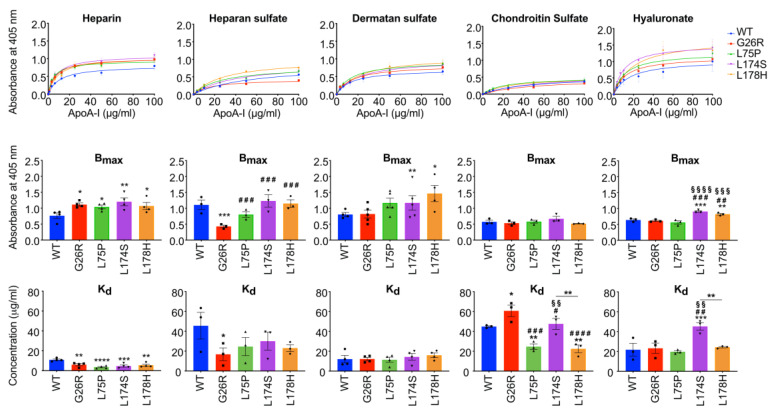
ApoA-I amyloidogenic variants’ differential binding to ECM components. 96-well plates were coated with 25 μg/mL of each ECM component under test and probed with ApoA-I amyloidogenic variants at increasing concentrations. The ability of variants to bind ECM components was quantified as a function of protein concentration (upper panels). The experimental data were fitted and B_max_ (middle panels) and K_d_ (lower panels) were calculated according to the Michaelis-Menten equation. Data shown are the means ± SEM of 3 to 5 independent experiments carried out in duplicate. Significance is calculated according to one-way ANOVA (* *p* < 0.05, ** *p* < 0.005, *** *p* < 0.001, **** *p* < 0.0001 for groups as shown with respect to the WT protein, # *p* < 0.05, ## *p* < 0.005, ### *p* < 0.001, #### *p* < 0.0001 for groups as shown with respect to G26R variant, §§ *p* < 0.005, §§§ *p* < 0.001, §§§§ *p* < 0.0001 for groups as shown with respect to L75P variant).

**Figure 5 ijms-24-00318-f005:**
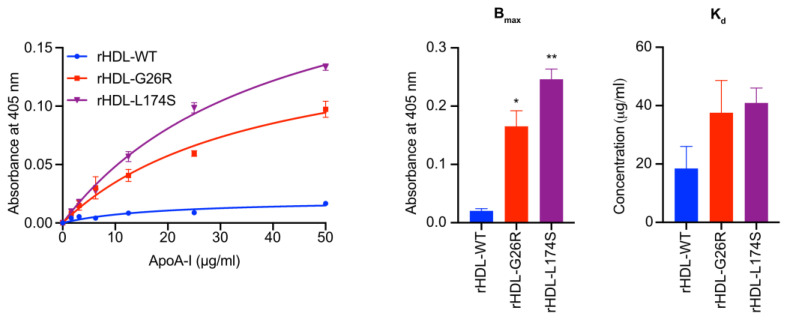
Lipidation decreases the binding of ApoA-I proteins to heparin. 96-well plates were coated with heparin at 25 ug/mL and probed with the rHDL containing the indicated ApoA-I amyloidogenic variants at increasing concentrations. The rHDL ability to bind heparin was quantified as a function of protein concentration (left panel). Experimental data were fitted according to the Michaelis-Menten equation and B_max_ (middle panel) and K_d_ (right panel) were calculated. Data shown are the means ± SEM of two independent experiments carried out in duplicate. Significance is calculated according to one-way ANOVA (* *p* < 0.05, ** *p* < 0.005).

**Figure 6 ijms-24-00318-f006:**
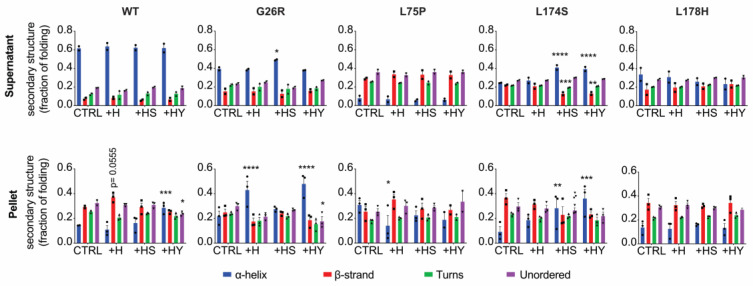
Secondary structure estimation of ApoA-I amyloidogenic variants in the presence of the different ECM components at pathophysiological pH. ApoA-I proteins (2 mg/mL) were incubated with the different ECM components (1:2 protein to ECM molar ratio) for one weeks in McIlvain buffer, pH 6.4, under agitation. At the end of the incubation, samples were centrifuged at 20,000× *g* for 30 min at 4 °C and both the supernatant and the insoluble material were analyzed by SRCD. CDSSTR algorithm was used to estimate the secondary structure distribution of ApoA-I proteins in the supernatant (upper panels) and insoluble fraction (lower panels). H = heparin, HS = heparan sulfate, HY = hyaluronate. Significance was calculated according to two-way ANOVA (* *p* < 0.05, ** *p* < 0.005, *** *p* < 0.001, **** *p* < 0.0001 for groups as shown with respect to the control).

**Figure 7 ijms-24-00318-f007:**
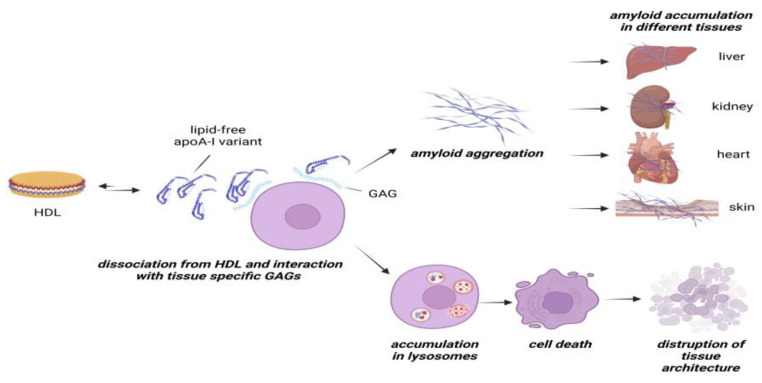
Proposed mechanism behind ApoA-I amyloidogenic variants cytotoxicity and preferential organ accumulation. ApoA-I proteins in circulation exist in two forms in equilibrium, lipid-free and the preferred lipid-bound state (HDL). In the case of amyloidogenic variants (depicted as the blue protein), this equilibrium is shifted towards the less structurally stable lipid-free form. The interaction of the lipid-free variants with tissue-specific GAGs leads to two consequences: (1) it promotes protein misfolding and amyloid aggregation, eventually resulting in fibrils accumulation in that tissue; (2) it promotes protein internalization in target cells, where the misfolded variants are redirected to lysosome and hamper lysosomal functionality, resulting in cell death and consequent disruption of tissue architecture. The figure has been created with BioRender.com, accessed on 21 September 2022.

## Data Availability

The data presented in this study are available on request from the corresponding author.

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
