# Peer review of "The Apparent Organ-Specificity of Amyloidogenic ApoA-I Variants Is Linked to Tissue-Specific Extracellular Matrix Components"

_ijms, 2022, doi:10.3390/ijms24010318_

Round 1

Reviewer 1 Report

Del Giudice et collaborators described the influence of Apo AI amyloidogenic variants on the viability and redox status of different types of cells (hepatocytes, cardiomyocytes, keratinocytes, and kidney cells), components of the amyloidosis target organs. Moreover, the authors present a preferential interaction of Apo AI amyloidogenic variants with extracellular matrix components (heparin, heparan sulfate, dermatan sulfate, chondroitin sulfate, and hyaluronate). They conclude there is a link between the apparent organ specificity of Apo AI amyloidogenic variants and the tissue-specific distribution of extracellular matrix components. The topic of this manuscript is novel and represents the current concerns in the field. The results are well presented, very clear, and appropriate. The manuscript could be accepted for publication in the International Journal of Molecular Science (in the special issue of“Molecular Research on Amyloidosis”) after some minor changes answering the comments below.

#1: The authors should present in the “Introduction” the current knowledge on the amino acid sequence of Apo AI protein and its mutations contributing to amyloidosis.  

#2: If it is possible, the authors should increase the resolution of confocal microscopy images, and should notice the magnification or scale bars of these images in figures 2 and 3.

#3:  The authors should explain why they choose the 2 µM concentration for each of the amyloidogenic variants of ApoAI to determine the influence of these proteins on target cells.

#4: At the line 538, 4.8.1 should be replaced with 4.8.2.

Reviewer 2 Report

Comments on “The apparent organ-specificity of amyloidogenic ApoA-I variants is linked to tissue-specific extracellular matrix components” from Del Giudice et al

The paper is of relevance as it describes physicochemical features (cytotoxicity, endocytic processing, conformation) in different conditions (focusing on ECM components) of several ApoA-I variants that are well known to produce amyloidosis. It is structured and clearly written. However, there are some aspects that need clarification.

I have a couple of questions and suggestions for the authors:

-      - If testing human ApoA-I variants, is there a rationale for using rat kidney and heart cells? If not, the authors should perform the MTT and confocal microscopy experiments also in human kidney and heart cells (as they did for skin and liver) as the results may differ.

-        - When performing the MTT assays with lipidated-ApoA-I variants the authors state that “No variant-specific cytotoxic effect was observed” but the tendency is the same as with lipid-free ApoA-I variants at 72h: G26R, L75P reduce viability in kidney cells and L178H reduces viability in skin cells. Indeed, the effects are milder but it seems to me that there is certain specificity variant-cell line. The authors may evaluate to rewrite this sentence (Line 120-121).

-       - The reviewer wonders why oxidative stress is induced with Adriamycin A in liver and kidney cells and with TBHP in skin and heart cells. Please provide a rational for not using the same for all or both each time.

-  - In results, at the “Cell-specific impairment of degradation of ApoA-I amyloidogenic variants” section the authors use H9c2 and NRK-52E cell lines to measure the uptake and intracellular levels of the ApoA-I variants. Why they only use these cell lines instead of the other two since these are the ones derived from rat?

-  - Regarding the results obtained using confocal microscopy, the overlap between ApoA-I and lysotracker seems not clear to me.  It is my opinion that the ApoA-I deposits observed in some conditions may be extracellular accumulations of the ApoA-I added to the medium (even in the wash-out condition) which could be consistent with the amyloidogenic properties of some of the variants. The authors should provide Z-Stack images at least, to demonstrate that the added ApoA-I is endocyted by the cells and further processed towards the lysosome where some forms of ApoA-I would accumulate. Additionally, the authors should conduct additional experiments using endocytosis inhibitors.

Minor comments

-          In Figure 4, it would be more visual if the authors used the same scale at the different graphics.

-          Spell check is required in some parts of the paper.
